# Anti-Osteoporotic Potential of Water Extract of *Anethum graveolens* L. Seeds

**DOI:** 10.3390/nu15194302

**Published:** 2023-10-09

**Authors:** Seon-A Jang, Sung-Ju Lee, Youn-Hwan Hwang, Hyunil Ha

**Affiliations:** 1Future Technology Research Center, KT&G Corporation, 30, Gajeong-ro, Yuseong-gu, Daejeon 34128, Republic of Korea; jsa85@ktng.com; 2KM Convergence Research Division, Korea Institute of Oriental Medicine, Yuseong-daero 1672, Daejeon 34054, Republic of Korea; sungjulee@kiom.re.kr (S.-J.L.); hyhhwang@kiom.re.kr (Y.-H.H.)

**Keywords:** *Anethum graveolens* L., osteoclastogenesis, osteoporosis, RANKL, ovariectomy

## Abstract

*Anethum graveolens* L., known as European dill, is a versatile herb widely used in both traditional medicine and culinary practices. Despite its long-standing history, the potential impact of the water extract of *A. graveolens* seeds (WEAG) on bone health remains unexplored. In this study, we investigated the influence of WEAG on osteoclast differentiation and assessed its potential as an anti-osteoporotic agent. WEAG hindered osteoclast differentiation through the suppression of receptor activator of nuclear factor-κB ligand (RANKL) expression in osteoclast-supporting cells and by directly targeting osteoclast precursor cells. WEAG significantly reduced the expression of key osteoclastogenic transcription factors, namely c-Fos and NFATc1, typically induced by RANKL in osteoclast precursors. This reduction was attributed to the suppression of both MAPKs and NF-κB pathways in response to RANKL. In vivo experiments further revealed that WEAG administration effectively reduces trabecular bone loss and weight gain triggered by ovariectomy, mimicking postmenopausal osteoporosis. Furthermore, our comprehensive phytochemical analysis of WEAG identified a range of phytochemical constituents, associated with bone health and weight regulation. Notably, we discovered a specific compound, isorhamnetin-3-O-glucuronide, within WEAG that exhibits anti-osteoclastogenic potential. Overall, this research elucidated the beneficial effects and mechanistic basis of WEAG on osteoclast differentiation and bone loss, indicating its potential as a viable alternative to address bone loss in conditions like postmenopause.

## 1. Introduction

Osteoporosis, a metabolic disorder, is characterized by diminished bone mass and microarchitectural deterioration, leading to weakened bone strength and an elevated susceptibility to fractures [1]. The process of bone remodeling is crucial for replacing old and damaged bone with new bone through the cooperation of osteoclasts and osteoblasts. During normal bone remodeling, the creation of new bone is intricately linked with the removal of old bone to ensure the maintenance of bone mass and quality. However, in various pathological conditions, bone resorption outpaces bone formation, resulting in excessive skeletal deterioration. One of the most common bone disorders, postmenopausal osteoporosis, arises from an increase in the formation and activity of osteoclasts due to the decrease in estrogen levels [2,3]. Heightened osteoclast formation and function are also associated with bone erosion in conditions such as rheumatoid arthritis and osteolysis seen in the osteolytic complications of metastatic tumors [4,5]. Osteoclasts, multinucleated giant cells, are derived from precursor cells within the monocyte/macrophage lineage through a series of processes including commitment, differentiation, fusion, and activation. The receptor activator of nuclear factor-κB ligand (RANKL) is a crucial cytokine that activates all phases of osteoclast development. RANKL is primarily synthesized by mesenchymal linage cells in bone tissue, with a notable source being osteocytes residing within the bone matrix [6,7]. When RANKL binds to RANK, it recruits tumor necrosis factor receptor (TNFR)-associated factor 6 (TRAF6) by initiating RANK trimerization. This initiation sets off a signaling cascade involving nuclear factor-κB (NF-κB) and mitogen-activated protein kinases (MAPKs), including p38, c-Jun N-terminal kinase (JNK), and extracellular signal-regulated kinase 1/2 (ERK1/2) [8]. Ultimately, the activation of these signaling molecules upregulates critical regulators like c-Fos and nuclear factor of activated T cell c1 (NFATc1), which directly oversee osteoclast differentiation and the expression of specific genes associated with osteoclastogenesis, such as dendritic cell-specific transmembrane proteins (DC-STAMP), ATPase H+ transporting V0 subunit D2 (Atp6v0d2), and cathepsin K (Ctsk) [9,10]. Therefore, targeting these signaling pathways holds promise as a strategy to inhibit osteoclastogenesis and mitigate osteoclast-mediated bone loss.

*Anethum graveolens* L. (*A. graveolens*), commonly referred to as European dill and belonging to the Umbelliferae family, is native to Europe, central and southern Asia, the Mediterranean region, and the southeastern part of Iran [11]. This plant boasts a rich history of traditional usage across various cultures, serving both culinary and medicinal purposes. Its seeds are used to add flavor as a spice, while its leaves are employed both as condiments and for the preparation of tea. Furthermore, dill oil extracted from the seeds, leaves, and stems is utilized as a flavor enhancer in the food industry. In traditional medicinal practices, it is primarily used to address digestive issues, including infant colic [12]. Previous research has revealed that the essential oil, aqueous, and alcoholic extracts of *A. graveolens* seeds exhibit a diverse range of pharmacological properties, including anti-inflammatory, antimicrobial, hypolipidemic, anti-diabetic, as well as gastric mucosal protective and antisecretory effects [11,13].

However, the impact of *A. graveolens* on bone health has yet to be determined. This study aims to explore the effects and potential mechanisms of the water extract of *A. graveolens* seeds (WEAG) on in vitro osteoclast differentiation and its bone-protective properties in an in vivo model of postmenopausal osteoporosis, utilizing ovariectomized (OVX) mice.

## 2. Materials and Methods

### 2.1. Materials

Antibodies targeting c-Fos (cat. no. sc-7202) and NFATc1 (cat. no. sc-7294) were purchased from Santa Cruz Biotechnology (Santa Cruz, CA, USA). Antibodies targeting phospho-I*κ*B*α* (cat. no. 2859S), I*κ*B*α* (cat. no. 9242), phospho-JNK1/2 (cat. no.9251), JNK (cat. no.9252), phospho-ERK1/2 (cat. no.9101), ERK (cat. no.9102), phospho-p38 (cat. no.9211), p38 (cat. no.9212), and β-actin (cat. no. 3700) were purchased from Cell Singling Technology (Danvers, MA, USA). Fast red violet LB Salt (cat. no. F3381), naphthol AS-MX phosphate (cat. no. N4875), and 4-Nitrophenyl phosphate disodium salt hexahydrate (cat. no. N2765) were purchased from Sigma-Aldrich (St. Louis, MO, USA).

### 2.2. WEAG Preparation

*A. graveolens* L. seeds were ground, subjected to heat reflux extraction with distilled water for 3 h, filtered, and then lyophilized. The obtained WEAG power was dissolved in distilled water and centrifuged at 10,000× *g* for 5 min, prior to its utilization in the experiments.

### 2.3. Bone Marrow-Derived Macrophages (BMMs) and MLO-Y4 Cell Culture

BMMs were cultured in α-MEM medium with 10% fetal bovine serum (FBS), 1% penicillin/streptomycin, and macrophage colony-stimulating factor (M-CSF) as previously described [14]. Osteocyte-like cell line, MLO-Y4 cells (Kerafast, Boston, MA, USA), were maintained in α-MEM with 2.5% FBS, 2.5% calf serum, and 1% penicillin/streptomycin in collagen-coated plates, following established cell culture maintenance protocols. Typical dendritic morphology, characteristic of these cells, was observed during culture maintenance.

### 2.4. Cell Viability Assay

Cells were seeded into a 96-well plate at the density of 2 × 10^4^ cells/well. The medium was removed after cells became adherent, and cells were treated with or without WEAG and cultured for 24 h. Cell viability assessments were carried out by incubating the cells with a Cell Counting Kit-8 reagent (CCK-8; Dojindo Molecular Technologies Inc., Rockville, MD, USA) for 2 h.

### 2.5. Osteoclast Differentiation

In order to induce osteoclast differentiation within an osteocyte-BMMs co-culture system, MLO-Y4 osteocyte cells (1 × 10^3^ cells/well) were initially seeded into 96-well plates, followed by the addition of BMMs (4 × 10^4^ cells/well) on the subsequent day. The co-cultures were treated with WEAG along with 1α,25-dihydroxyvitamin D3 (VitD3; 10 nM) or VitD3 and RANKL (50 ng/mL) for a duration of 5 days. To stimulate osteoclast differentiation in BMMs, BMMs (1 × 10^4^ cells/well) were treated with WEAG or with phytochemicals identified in WEAG in the presence of RANKL (50 ng/mL) and M-CSF (60 ng/mL) in 96-well plates for 4 days. All cultures were substituted with fresh medium together with supplements on day 3.

### 2.6. Tartrate-Resistant Acid Phosphatase (TRAP) Activity and Staining

Cells were fixed and subsequently permeabilized with 0.1% Triton X-100. Next, the cells were incubated with TRAP solution at 37 °C for 20 min. The resulting supernatant was combined with 0.1 N NaOH in a 1:1 ratio, and absorbance was measured at 405 nm. Following TRAP activity measurement, the cells were stained to a violet color using a solution comparing fast red violet LB salt and naphthol AS-MX phosphate.

### 2.7. Western Blot Analysis

Cultured cells were lysed in ice-cold protein lysis buffer containing protease and phosphatase inhibitors and then cleared via centrifugation. Equivalent amounts of extracted proteins were separated using 10% SDS–PAGE and then transferred to the PVDF membranes (Millipore, MA, USA). After being blocked at room temperature in 5% skim-milk for 1 h, the membranes were incubated overnight at 4 °C with the primary antibodies. After washing with TBS-T buffer, the membranes were incubated with HRP-conjugated secondary antibody for 1 h. The target bands were detected using SuperSignal^®^ West Pico Chemiluminescent Substrate.

### 2.8. Quantitative Real-Time Polymerase Chain Reaction (PCR)

Total RNA was extracted on the indicated days using the RNA-spin^TM^ Total RNA Extraction Kit (iNtRON Biotechnology, Sungnam, Republic of Korea), and purified total RNA was used for cDNA synthesis with the High-Capacity cDNA Reverse Transcription Kit (Thermo Fisher Scientific, Waltham, MA, USA). The target mRNAs were quantified using the TaqMan Universal Master Mix II (Applied Biosystems, Foster City, CA, USA) along with TaqMan probes on an ABI 7500 Real-Time PCR system (Applied Biosystems). Data analysis was conducted following the comparative cycle threshold method [15], with normalization to 18S expression in each sample.

### 2.9. Animal Experiment

Six-week-old female C57BL/6J mice (Central Lab. Animal Inc., Seoul, Republic of Korea) were acclimatized under controlled laboratory conditions. After a one-week acclimatization period, bilateral OVX was performed to induce an osteoporosis model. The sham control group consisted of mice whose ovaries were not removed. Previous studies have shown that the oral administration of an aqueous extract of *A. graveolens* seeds resulted in a dose-dependent reduction in HCl-induced gastric lesions in mice, with effective doses ranging from 50 to 450 mg/kg [16]. Building upon these findings, we opted to investigate the impact of WEAG on bone loss in OVX mice using doses of 30 mg/kg/day (referred to as WEAG-L) and 100 mg/kg/day (referred to as WEAG-H). One week after surgery, we established the following four groups (*n* = 6 each): sham, OVX, OVX treated with WEAG at 30 mg/kg/day (WEAG-L), and OVX treated with WEAG at 100 mg/kg/day (WEAG-H). The mice were provided with a normal-fat diet (10 kcal%; Research Diet, New Brunswick, NJ, USA) and had access to water ad libitum, while also receiving daily oral gavage administration of WEAG for a duration of 5 weeks.

### 2.10. Micro-Computed Tomography (μ-CT) Analysis

For the analysis of trabecular bone microstructure, we employed the micro-CT SkyScan 1276 system (Bruker, Kontich, Belgium) to perform scans on the distal femoral bone. The reconstruction of the scanned images was carried out using the NRecon software (version 1.7.42, Bruker). Three-dimensional images of the distal femoral bone were generated using CTvol software (version 2.3.2.0, Bruker). The volume of interest for trabecular bone analysis in the distal femur spanned from 50 µm below the growth plate to the proximal direction, covering a total length of 1.0 mm. Bone mineral density (BMD) and bone morphometric parameters were analyzed using CTAn software (version 1.20.3.0, Bruker).

### 2.11. Ultrahigh-Performance Liquid Chromatography–Tandem Mass Spectrometry (UHPLC-MS/MS) Analysis

Reference standard phytochemicals, including quinic acid, 5-O-caffeoylquinic acid, caffeic acid, schaftoside, feruloylquinic acid, and quercetin-3-O-rutinoside, were purchased from Targetmol (Wellesley Hills, MA, USA), and quercetin-O-glucuronide, kaempferol-3-O-glucuronide, and isorhamnetin-3-O-glucuronide were obtained from ChemFace (Wuhan, China). WEAG were analyzed using a Dionex UltiMate 3000 system equipped with a Thermo Q-Exactive mass spectrometer, following previously published protocols with some modifications [17,18]. Chromatographic separation was performed using a Dionex UltiMate 3000 system coupled with a C18 column (Acquity BEH, 100 × 2.1 mm, 1.7 μm) and a gradient system comprising 0.1% formic acid in water and acetonitrile. The Q-Exactive mass spectrometer operated in negative ion mode with a heated electrospray ionization source. The acquired data were subsequently analyzed using Xcalibur software (version 4.1, Thermo Fisher Scientific).

### 2.12. Statistical Analysis

In vivo data are presented as the mean ± standard error of the mean (SEM), while in vitro data are represented as mean ± standard deviation (SD). The values were assessed via one-way analysis of variance (ANOVA) followed by Dunnett’s post hoc test or two-way ANOVA followed by Sidak’s post hoc test.

## 3. Results and Discussion

### 3.1. WEAG Inhibits Osteoclastogenesis In Vitro

Osteocytes are terminally differentiated cells of the osteoblastic lineage, distributed throughout the mineralized bone matrix [19]. They regulate the activities of osteoclasts and osteoblasts, making them pivotal in bone remodeling. Recent research has unveiled osteocytes as a primary source of RANKL [6,20]. The osteocyte-like cell line, MLO-Y4, which exhibits many characteristics of osteocytes [21], expresses high levels of RANKL and can support osteoclast formation when co-cultured with osteoclast precursor cells [22]. To assess the influence of WEAG on osteoclast formation in vitro, our initial investigation focused on its effect in a co-culture system comprising MLO-Y4 osteocyte-like cells and BMMs. Consistent with previous reports [23], VitD3 stimulated osteoclast formation in the co-culture, as evidenced by the presence of TRAP-positive multinucleated cells containing more than three nuclei. The inhibitory effect of WEAG on osteoclast formation was observed in a dose-dependent manner (Figure 1A). Notably, even when exogenous RANKL was introduced under identical co-culture conditions, WEAG continued to exert its inhibitory effect, implying that WEAG inhibits osteoclast formation independently of RANKL production (Figure 1A). Supporting these findings, we additionally validated that WEAG exhibited a dose-dependent reduction in RANKL-induced osteoclast differentiation in isolated BMMs (Figure 1B). Importantly, WEAG treatment did not induce cytotoxic effects on BMMs, regardless of the concentration used (Figure 1C). Next, we explored whether WEAG has any influence on the expression of RANKL and the decoy receptor osteoprotegerin (OPG). The treatment of MLO-Y4 cells with VitD3 significantly increased RANKL mRNA levels while decreasing OPG mRNA levels. In contrast, WEAG inhibited VitD3-stimulated RANKL mRNA expression but did not influence OPG mRNA expression (Figure 1D). These results collectively suggest that WEAG is capable of inhibiting osteoclast formation by directly targeting osteoclast precursors and simultaneously suppressing RANKL expression in osteoclast-supporting MLO-Y4 cells.

### 3.2. WEAG Inhibits RANKL-Mediated Signaling Pathways

To elucidate the molecular mechanisms underpinning WEAG’s inhibitory effect on osteoclast precursor differentiation, we conducted a thorough examination of WEAG’s impact on the expression and activation of RANKL-induced signaling pathways central to osteoclastogenesis. Upon RANKL binding to RANK, it triggers the activation of NF-κB and MAPKs via TRAF6, culminating in the upregulation of c-Fos and NFATc1. During the osteoclastogenesis process, the heightened activity of NFATc1 serves to activate and induce the expression of osteoclast-specific genes encoding vital proteins necessary for osteoclast differentiation, fusion, and function. Atp6v0d2 and DC-STAMP play crucial roles in mediating osteoclast cell–cell fusion, while CtsK is intimately involved in bone resorption [24]. Concurrently, negative regulators that impede the transcriptional activity of NFATc1, such as interferon regulatory factor-8 (IRF-8) and v-maf musculoaponeurotic fibrosarcoma oncogene family protein B (MafB), are downregulated by the transcriptional repressor B-lymphocyte-induced maturation protein 1 (Blimp1) during the course of osteoclast differentiation [25,26,27]. The treatment of BMMs with WEAG produced significant inhibition in RANKL-induced mRNA and the protein expression of c-Fos and NFATc1 (Figure 2A,B). Subsequently, WEAG also effectively dampened the mRNA expression of osteoclast-specific genes, including Atp6v0d2, DC-STAMP, and CtsK (Figure 2B). Furthermore, WEAG hindered the reduction in the expression of IRF-8 and MafB, while simultaneously inhibiting the increase in Blimp1 expression observed during RANKL-induced osteoclastogenesis. These results collectively suggest that WEAG impedes osteoclast differentiation by concurrently inhibiting the upregulation of positive regulators and the downregulation of negative regulators crucial for osteoclastogenesis. Prior research has emphasized the critical role of p38 in osteoclastogenesis, as it is associated with RANKL-induced c-Fos and NFATc1 expression, and it facilitates CtsK expression through its interaction with NFATc1 [28,29]. Additionally, JNK activation and its downstream target c-Jun have been implicated in RANKL-induced c-Fos and NFATc1 expression in osteoclast precursors [30,31]. Conversely, the role of ERK in regulating osteoclast differentiation appears to be context- and stage-dependent [32]. In addition to the MAPK pathways, the NF-κB signaling pathway, a critical participant in inflammatory and immune responses, plays a central role downstream of RANKL in osteoclastogenesis [33,34]. In the inactive state, NF-κB forms a complex with the NF-κB inhibitor IκBα, which impedes the translocation of NF-κB, consisting of the p65 and p50 subunits, into the nucleus. Upon stimulation by RANKL, IKK triggers the phosphorylation and subsequent degradation of IκBα, releasing NF-κB for nuclear translocation [33]. To delve deeper into WEAG’s inhibitory effect on NFATc1 expression and osteoclastogenesis, we investigated whether WEAG influences the activation of RANKL-induced MAPKs and NF-κB. Significantly, WEAG treatment exhibited the inhibition of the RANKL-induced phosphorylation of JNK and p38, with no discernible impact on ERK phosphorylation. Furthermore, it suppressed RANKL-induced NF-κB activation, as evidenced by the degradation of IκBα (Figure 2C). Our findings collectively suggest that WEAG’s anti-osteoclastogenic effect can be attributed to its ability to negatively regulate the induction of NFATc1 and the repressors IRF-8 and MafB by inhibiting the RANKL-induced activation of the JNK, p38, and NF-κB signaling pathways.

### 3.3. WEAG Inhibits Bone Loss

Given the observed inhibitory effect of WEAG on osteoclast differentiation, we examined the potential pharmacological impact of WEAG using an OVX mouse model. This model induces estrogen deficiency-induced osteoporosis, leading to an increase in RANKL expression that elevates bone resorption and bone turnover rate. Ultimately, it leads to a reduction in BMD and significant disruption of the trabecular microstructure [35]. μ-CT technology offers the capacity to provide both quantitative and visually informative data concerning bone microarchitecture. Three-dimensional μ-CT images provide a clear visualization of the trabecular bone and enable the acquisition of quantitative parameters, encompassing BMD, bone volume per tissue volume (BV/TV), trabecular number (Tb.N), trabecular thickness (Tb.Th), and trabecular separation (Tb.Sp).

After 6 weeks of ovariectomy, μ-CT analysis of the femoral bone showed a notable reduction in trabecular bone in the OVX group when compared to the sham group. However, both WEAG-L (30 mg/kg/day) and WEAG-H (100 mg/kg/day) treatments effectively mitigated this bone loss. WEAG administration at both doses exhibited increased BMD, BV/TV, Tb.N, and Tb.Th, but decreased Tb.Sp compared to the OVX group (Figure 3A,B). During the progression of osteoporosis induced by OVX surgery, uterine weight notably decreases while body weight increases [36]. In line with this, our study identified a substantial increase in body weight in the OVX group compared to the sham group. Remarkably, the elevated body weights observed in the OVX group were effectively restored to normal by administering two different doses of WEAG. However, in contrast to body weight changes, uterine atrophy induced by ovariectomy was not reversed in the WEAG-treated groups (Figure 3C).

Our animal study collectively demonstrates that WEAG effectively mitigates bone loss and suppresses body weight gain induced by ovariectomy, without exerting estrogenic effects, as evidenced by the absence of an impact on uterine atrophy. While further investigation is warranted, including an in-depth exploration of WEAG’s influence on osteoclasts and osteoblasts in vivo, to fully uncover the underlying mechanisms responsible for the beneficial effects of WEAG observed in OVX mice, our findings hold significant implications for the potential use of WEAG as a therapeutic candidate for addressing two major health concerns prevalent in postmenopausal women—osteoporosis and obesity.

### 3.4. Phytochemical Profiling of WEAG

UHPLC-MS/MS analysis stands as a leading analytical method, providing accurate insight into the active components present within plant samples. This method facilitates detailed chemical profiling even before proceeding to the isolation and purification stages [37]. To uncover the molecular foundation behind WEAG’s biological activity, we delved into its phytochemical composition. Through UHPLC-MS/MS, we discerned six phenolics (quinic acid, 3-O-caffeoylquinic acid, 5-O-caffeoylquinic acid, caffeic acid, feruloylquinic acid, malonyl-tri-O-caffeoylquinic acid) and five flavonoids (schaftoside, quercetin-3-O-rutinoside, quercetin-O-glucuronide, kaempferol-3-O-glucuronide, iso-rhamnetin-3-O-glucuronide) present in WEAG, as detailed in Table 1. Figure 4 displays both the base peak chromatograms and the extracted ion chromatogram of the pinpointed phytochemicals in WEAG.

### 3.5. The Effects of Phytochemicals in WEAG on Osteoclast Differentiation

We proceeded to examine the influence of the phytochemicals identified in WEAG on osteoclast differentiation, aiming to determine if the inhibitory effect of WEAG on osteoclastogenesis could be ascribed to the bioactive properties of these constituent phytochemicals. Out of the eleven identified compounds, our investigation centered on nine readily available phytochemicals. Significantly, we discovered that one particular compound, isorhamnetin-3-glucuronide, potently suppressed RANKL-induced osteoclast differentiation without any associated cytotoxicity (Figure 5A,B). While the effects of the phytochemicals in WEAG on bone metabolism remain underexplored, it is important to note that caffeic acid has displayed varied impacts on bone—both beneficial and harmful—depending on the specific animal model used [38]. Isorhamnetin-3-O-glucuronide, which showed anti-osteoclastogenic effects in our study, has previously demonstrated anti-inflammatory properties by inhibiting LPS-stimulated JNK and p38 activation in RAW264.7 macrophage cells [39]. In alignment with our observations on isorhamnetin-3-O-glucuronide, its aglycone variant, isorhamnetin, has been found to counteract RANKL-induced osteoclastogenesis in BMMs. Moreover, in a surgically induced osteoarthritis mouse model, the administration of isorhamnetin resulted in a reduction in osteoclast numbers and a decrease in bone loss within the subchondral region [40].

Regarding WEAG’s potential anti-obesity activity, several of its phytochemicals, including quinic acid [41], 5-O-caffeoylquinic acid [42], caffeic acid [43], and kaempferol-3-O-glucuronide [44], have exhibited anti-obesity effects in various animal studies.

The unearthing of isorhamnetin-3-O-glucuronide’s anti-osteoclastogenic ability, combined with the detection of phytochemicals with anti-obesity properties within WEAG, offers profound insights into the underlying mechanisms of WEAG’s dual action against osteoporosis and obesity. This suggests the potential of WEAG as a comprehensive alternative treatment, addressing both bone health and weight concerns, which are critical health challenges often faced by postmenopausal women.

## 4. Conclusions

This research illuminates the previously uncharted potential of WEAG for bone protection. We have shown that WEAG curbs osteoclast differentiation by dampening RANKL expression in osteoclast-supportive cells and directly intervening in RANKL-induced osteoclastogenesis. Additionally, administering WEAG to OVX mice markedly reduced both bone loss and weight gain. Through our phytochemical profiling of WEAG, we pinpointed several constituents recognized for their influence on bone health and obesity management. Notably, we discovered a particular compound, isorhamnetin-3-O-glucuronide, that exhibits an anti-osteoclastogenic effect. Collectively, these revelations indicate that WEAG could be a promising pharmacological solution to counter bone loss stemming from heightened osteoclast formation. This potential is especially pertinent when considering conditions like postmenopausal osteoporosis, which often coexists with weight management challenges.

## Figures and Tables

**Figure 1 nutrients-15-04302-f001:**
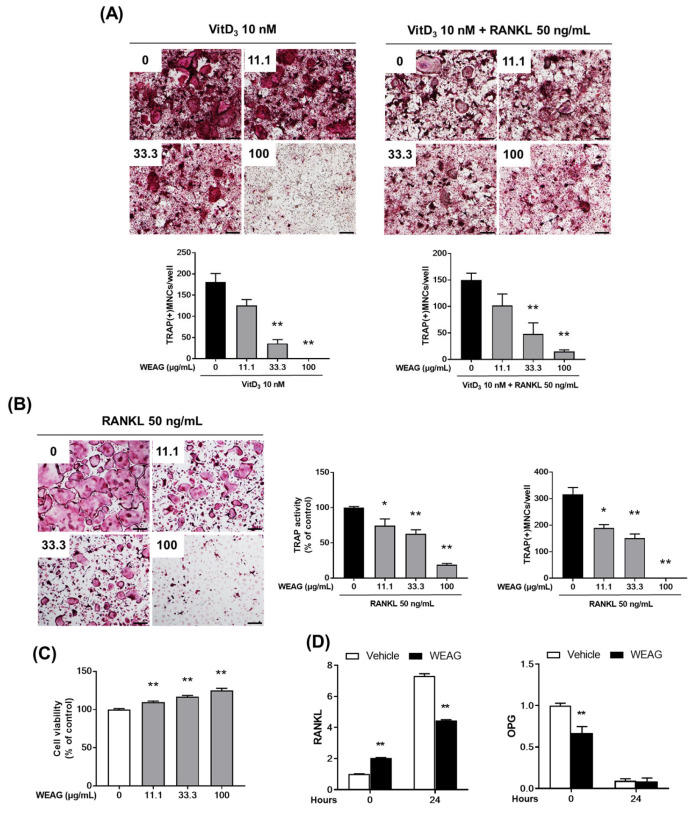
WEAG inhibits osteoclast differentiation. (**A**) BMMs and MLO-Y4 cells were co-cultured with or without WEAG (11.1, 33.3 and 100 µg/mL) in the presence of VitD3 (10 nM) or VitD3 (10 nM) plus RANKL (50 ng/mL) for 5 days. TRAP-stained images (**upper panel**; scale bar, 200 µm) and the number of TRAP-positive multinucleated cells (MNCs; **lower panel**) containing more than three nuclei (*n* = 3/group, except for 0, *n* = 9). (**B**) BMMs were treated with or without WEAG (11.1, 33.3 and 100 µg/mL) in the presence of M-CSF (60 ng/mL) and RANKL (50 ng/mL) for 4 days. TRAP-stained images (**left panel**; scale bar, 200 µm), the cellular TRAP activity (**middle panel**), and the number of TRAP-positive MNCs (**right panel**); *n* = 3/group, except for 0, *n* = 9. (**C**) The cell viability of BMMs was assessed using a CCK-8 assay after 24 h of treatment with or without WEAG (11.1, 33.3, and 100 µg/mL) in the presence of M-CSF (60 ng/mL); *n* = 3/group, except for 0, *n* = 9. (**D**) MLO-Y4 cells were treated with or without WEAG (100 μg/mL) and VitD3 (10 nM) for 24 h. RANKL and OPG mRNA expression levels were quantified via real-time PCR (*n* = 3/group). * *p* < 0.05, ** *p* < 0.01 vs. control.

**Figure 2 nutrients-15-04302-f002:**
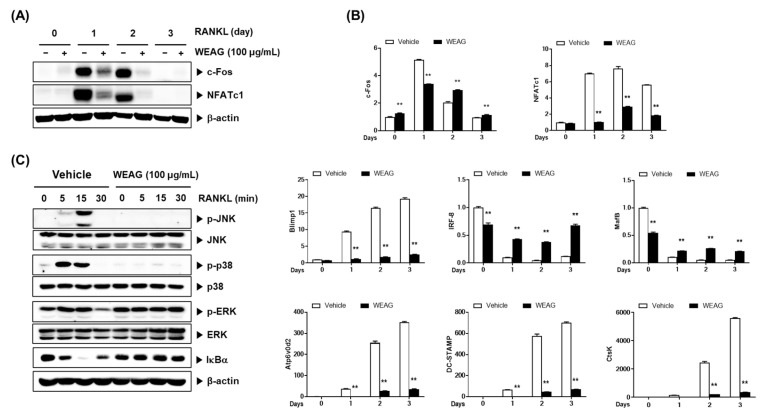
WEAG inhibits RANKL-induced signaling pathways in BMMs. (**A**,**B**) BMMs were treated with or without WEAG (100 µg/mL) and RANKL (50 ng/mL) for the indicated durations. Protein expression levels of the indicated proteins (**A**) and mRNA expression levels of the indicated genes (**B**) were assessed using Western blotting and real-time PCR (*n* = 3/group), respectively. (**C**) BMMs were stimulated with RANKL for the indicated time intervals following a 3 h pretreatment with WEAG. Protein levels of phosphorylated and non-phosphorylated forms were detected via Western blotting. ** *p* < 0.01 vs. control.

**Figure 3 nutrients-15-04302-f003:**
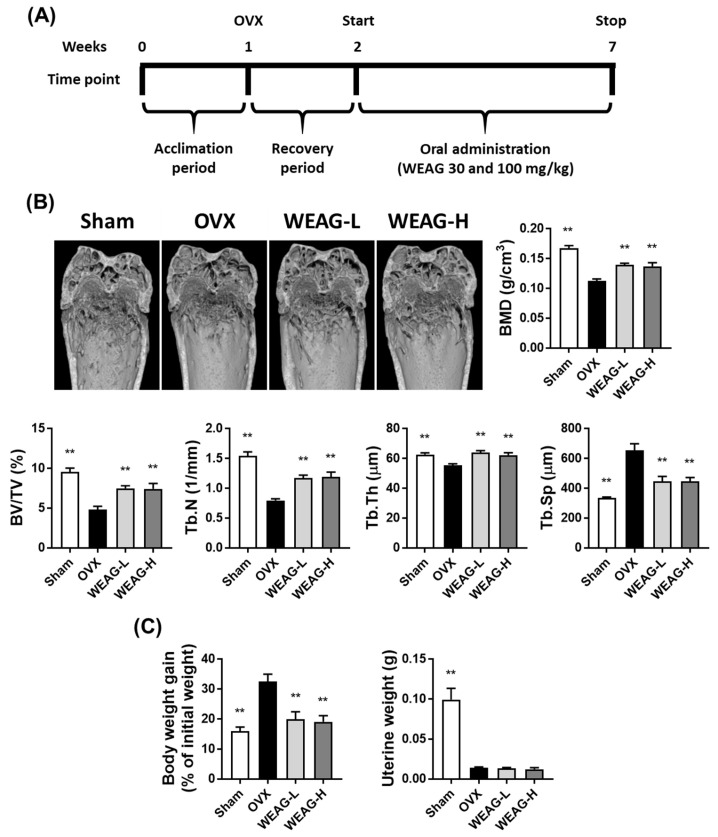
WEAG mitigates bone loss in OVX mice. (**A**) Schematic representation of the experimental protocols. (**B**,**C**) Sham-operated (sham) or OVX mice were orally administered with either vehicle, WEAG at a dose of 30 mg/kg (WEAG-L), or 100 mg/kg (WEAG-H) for 5 weeks. (**B**) µ-CT images of the distal femurs and subsequent evaluation of BMD and bone morphometric parameters in the femoral trabecular bone (*n* = 6/group). (**C**) Assessment of changes in body weight and uterine weight (*n* = 6/group). ** *p* < 0.01 vs. OVX group.

**Figure 4 nutrients-15-04302-f004:**
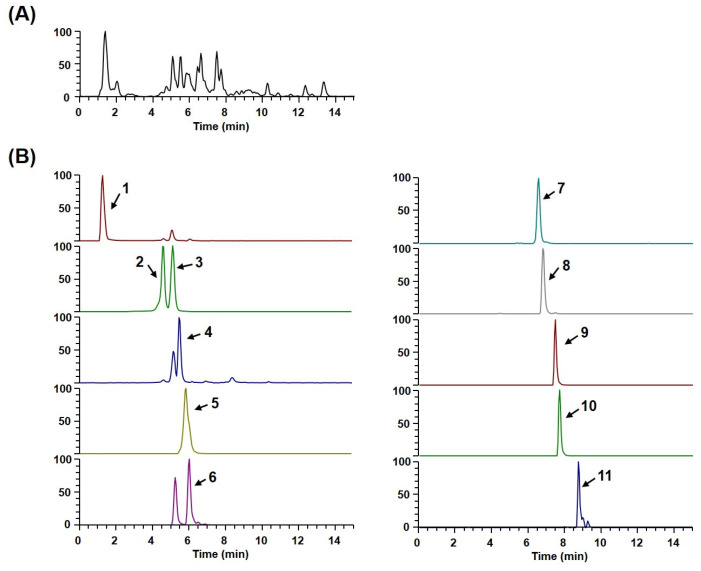
UHPLC-MS/MS analysis of WEAG. (**A**) Base peak chromatogram. (**B**) Extracted ion chromatogram of identified phytochemicals. 1, quinic acid; 2, 3-O-caffeoylquinic acid; 3, 5-O-caffeoylquinic acid; 4, caffeic acid; 5, schaftoside; 6, feruloylquinic acid; 7, quercetin-3-O-rutinoside; 8, quercetin-O-glucuronide; 9, kaempferol-3-O-glucuronide; 10, isorhamnetin-3-O-glucuronide; 11, malonyl-tri-O-caffeoylquinic acid.

**Figure 5 nutrients-15-04302-f005:**
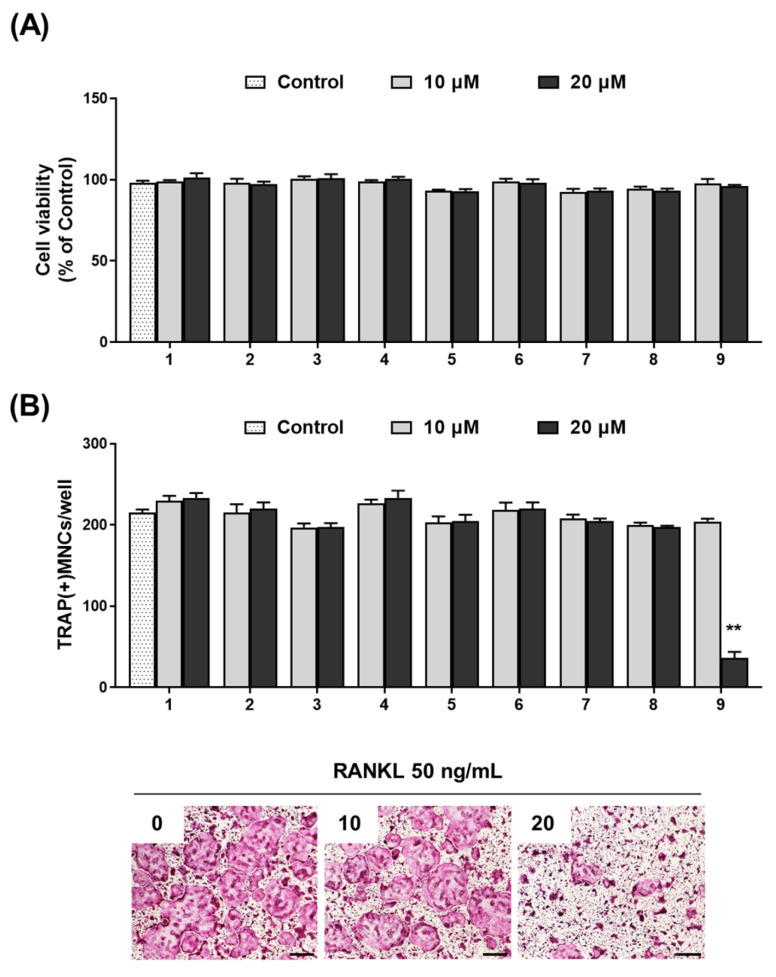
Impact of phytochemicals in WEAG on RANKL-induced osteoclast differentiation. (**A**) Cell viability of BMMs was assessed using a CCK-8 assay after 24 h of treatment with or without the phytochemicals (each 10 and 20 µM) in the presence of M-CSF (60 ng/mL); *n* = 3/group, except for 0, *n* = 9. (**B**) BMMs were treated with or without the phytochemicals (each 10 and 20 µM) in the presence of M-CSF and RANKL for 4 days. The number of TRAP(+) MNCs containing more than three nuclei (**upper panel**; *n* = 3/group) and representative images of TRAP staining treated with isorhamnetin-3-O-glucuronide (**lower panel**; scale bar, 200 µm). 1, quinic acid; 2, 5-O-caffeoylquinic acid; 3, caffeic acid; 4, schaftoside; 5, feruloylquinic acid; 6, quercetin-3-O-rutinoside; 7, quercetin-O-glucuronide; 8, kaempferol-3-O-glucuronide; 9, isorhamnetin-3-O-glucuronide. ** *p* < 0.01 vs. control.

**Table 1 nutrients-15-04302-t001:** Identification of phytochemicals in WEAG via UHPLC-MS/MS analysis.

No	t_R_ (min)	[M-H]^−^ (*m*/*z*)		Elemental Composition	Error (ppm)	MS/MS Fragments (*m*/*z*)	Identification
		Estimated	Calculated				
1	1.36	191.0561	191.0553	C_7_H_12_O_6_	1.80	173.045, 127.039, 85.028	Quinic acid *
2	4.67	353.0878	353.0879	C_16_H_18_O_9_	3.26	191.055, 179.034	3-O-caffeoylquinic acid*
3	5.22	353.0878	353.0879	C_16_H_18_O_9_	3.35	191.055, 179.034, 173.045	5-O-caffeoylquinic acid*
4	5.54	179.035	179.0341	C_9_H_8_O_4_	1.47	135.044	Caffeic acid *
5	5.95	563.1406	563.1407	C_26_H_28_O_14_	1.94	473.109, 443.098, 383.077	Schaftoside *
6	6.13	367.1035	367.1034	C_17_H_20_O_9_	2.65	193.05, 173.045	Feruloylquinic acid *
7	6.54	609.1461	609.1466	C_27_H_30_O_16_	2.62	427.103, 301.034	Quercetin-3-O-rutinoside *
8	6.77	477.0675	477.0676	C_21_H_18_O_13_	2.64	409.118, 301.035	Quercetin-O-glucuronide *
9	7.46	461.0725	461.0728	C_21_H_18_O_12_	2.89	285.04	Kaempferol-3-O-glucuronide *
10	7.69	491.0831	491.0833	C_22_H_20_O_13_	2.58	315.051	Isorhamnetin-3-O-glucuronide *
11	8.72	763.2455	763.2458	C_36_H_44_O_18_	1.91	179.034, 137.023	Malonyl-tri-O-caffeoylquinic acid

*, compared with an authentic standard.

## Data Availability

All data of this study are provided within this study.

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
