# Peer review of "Anti-Osteoporotic Potential of Water Extract of Anethum graveolens L. Seeds"

_nutrients, 2023, doi:10.3390/nu15194302_

Round 1
Reviewer 1 Report
This study aims to explore the effects and potential mechanisms of the water extract of A. grav-76 eolens seeds (WEAG) on in vitro osteoclast differentiation as well as its bone-protective properties in an in vivo model of postmenopausal osteoporosis, utilizing ovariectomized 78 (OVX) mice.
The methods and conclusions are sound. Still, the identification of osteocytes and osteoclast requires use of specific surface markers.
Reviewer 2 Report
Seon and colleagues present a comprehensive study that delves into the effects of the water extract of A. graveolens seeds (European dill) on osteoclast differentiation. Their findings indicate that this extract, particularly the compound isorhamnetin-3-O-glucuronide, may inhibit osteoclast differentiation and reduce bone loss in postmenopausal osteoporosis models, thereby highlighting potential therapeutic implications. While the overall premise of the study is both intriguing and relevant, there are several specific areas that require further clarification or expansion in order to strengthen the manuscript for publication:
1. Regarding the cellular effects of WEAG and isorhamnetin-3-O-glucuronide: Have the authors investigated if these compounds influence the viability of BMMs? An elucidation on this could help rule out potential cytotoxic effects, ensuring the results are indeed tied to differentiation pathways.
2. In the animal experiments, the authors have categorized doses of 30 mg/kg and 100 mg/kg for WEAG as low and high, respectively. It would be beneficial for readers to understand the rationale behind this categorization. Was this based on previous literature, pilot studies, or other considerations?
3. I noticed that the figure legends seem to lack explicit sample sizes, commonly referred to as 'n numbers'. This information is paramount for ensuring statistical validity and reproducibility. Including this would provide readers with a clearer understanding of the experimental design and robustness.
4. In relation to the animal experiments, and to further support the findings, it would be advisable to conduct TRAP staining. This would provide visual evidence and validation of the observed reductions in Oc.S/BS and N. Oc/Bs, ensuring that the reductions are not merely statistical anomalies.
5. An aspect worth exploring is the broader impact of WEAG on bone physiology. Specifically, it's imperative to determine whether WEAG might also influence bone formation. A preliminary in vitro study, at the very least, should be conducted to assess WEAG's effects on BMSCs osteoblast differentiation, thereby offering a holistic view of its potential therapeutic value.
The English quality of the paper is commendable, especially given the technical nature of the content. However, some adjustments can enhance clarity and readability.
Round 2
Reviewer 2 Report
Having carefully reviewed the revised version of the manuscript, I am pleased to note significant improvements over the initial submission. The authors have evidently made a commendable effort in addressing the concerns raised during the previous review, resulting in a more robust and coherent presentation of their findings. I recommend accepting the manuscript for publication.